# A hierarchical fusion framework to integrate random projectionbased classifiers: application in head and neck squamous carcinoma cancer

*Abstract:*Ensemble methods achieves better performance than single classifier model. Classifier diversity and fusion architecture are equally important for building a successful multi-classifier system. In this study, we introduced random projection to obtain the required classifier diversity and then proposed a hierarchical framework, namely a novel hierarchical fusion integrating random projection diversified classifiers (HFRPC). The proposed hierarchical fusion scheme was validated on survival prediction of head and neck squamous carcinoma cancer (HNSCC). Experimental results have demonstrated the superiority of the proposed HFRPC framework over the base classifier member and the state-of-the-art benchmark ensemble methods, rendering it a potential tool to assist medical decision making in the practical clinical setting.

*Keywords:* ensemble method, random projection, fusion architecture, ensemble diversity

## I.    Introduction

Ensemble methods are statistical and computational procedures that combine decisions from different learning algorithms (e.g. classifiers) to obtain more reliable and more accurate predictions than a single classifier in supervised and unsupervised learning problems [1]. Several theoretical bases demonstrated the effectiveness of an ensemble learning system[1]. The practical reasons for combining multiple classifiers are the successful applications of the ensemble methods in diverse fields.

The variety of ensemble methods proposed in recent years can be categorized into two types in the literature [2], heterogeneous and homogeneous ensemble. The heterogeneous ensemble methods underline the way on how to select or combine the existing base classifiers (usually generated in one training set) instead of emphasising how different and diverse classifiers are generated. In contrast, the homogeneous ensemble methods generate sets of base learners that attempt to improve diversity by manipulating the learning algorithm or structure of the dataset, e.g. a set of base classifiers is produced by applying a single learning algorithm to different training sets obtained from an original one. The aforementioned taxonomy of the ensemble methods characterises two important issues, i.e. fusion architecture and ensemble diversity, that need to be considered in the construction of a multi-classifier system (MCS).

An effective MCS is especially in demand in the medical decision-making context, which is necessary to deal with clinical tasks such as diagnosis, prognosis, and outcome prediction employing collected information from radiological images. The exigent needs for a potent MCS in clinics are mainly attributed to the following practical reasons. First, a medical decision is usually determined by considering various sources of information. In this regard, a fusion system is required to integrate the miscellaneous information to aid in making clinical decisions or determining treatment options. Secondly, medical-decision problems are often problem-specific, and various classifiers might have

distinct performance with respect to different diseases and clinical endpoints. Even for the same clinical task, different classifiers seldom reach unanimous results. Therefore, an efficient MCS framework is always expected in a clinical setting to tackle the comprehensive multifaceted medical data.

In the present study, we propose a novel random-projection-based ensemble framework where the original dataset is mapped to a lower dimensional space using a number of random-projection matrices to generate diversified training datasets. The proposed hierarchical fusion scheme is validated on survival prediction of HNSCC.

## II.  Methods and experiments

In HFRPC, we first use 10 random projection matrix generated by Eq.1 to transformed the original dataset into lower dimensional spaces, then each of the random projected samples was fed into several base classifiers, leading to the generation of a large amount of classifiers to be combined. Instead of directly fusing all these classifiers in a parallel manner, a two-level hierarchical framework was adopted. In level one, we essentially summarised the performance of each base classifier within the scope of the projection space. In level two, we quantified and weighed each base classifier and finally yielded the performance representation of each class. A two-level hierarchical fusion integrating random projection diversified multi-classifier (HFRPC) framework was devised and utilised to integrate all the outputs from the diversified multi-classifiers to yield a final prediction. Seven typical classifiers, namely, the Gaussian Bayes (GB), logistic regression (LR), quadratic discriminant analysis (QDA), $k$-nearest neighbour (KNN), decision tree (DeT), random forest (RF) [3], and XGBoost [4], were employed as the base classifiers for the proposed framework. We compared the proposed method with each of the base classifier in the classifier pool as well as with several state-of-the-art ensemble methods, including the plurality-voting (PV) method [5], conventional weighted-fusion (WF) method [6], stacking method [7], and DT [8, 9]. $\hat{x} \quad 1 x \quad \mathbf{P}_l$

$$(1) \quad \overline{\sqrt{q}}$$

## III.  Results and conclusions

The HNSCC dataset was collected from the Head–Neck-Radiomics-HN1 [10] in the Cancer Imaging Archive (TCIA, http://www.cancerimagingarchive.net). This dataset consists of clinical data and CT from 137 HNSCC patients treated with radiotherapy in which 74 (54%) patients died and 63 (46%) survived. For these patients, the 3D volume of the gross tumour volume (GTV) was manually delineated on the pre-treatment CT scan by an experienced radiation oncologist. We extracted the radiomic features using the contoured GTV delineated from the CT images and then concatenated them using the clinical features to a unitary feature that served as an input for a classification model. A five-time five-fold cross validation was implemented for performance evaluations. The classification performance was quantified using the ACC, the area under the receiver operating characteristic (ROC) curve (AUC), sensitivity (SEN), and specificity (SPE). The proposed HFRPC achieved the highest ACC and AUC than all the base classifiers and all benchmark ensemble methods, whereas the highest SEN and SPE were observed in base classifiers XGBoost and Stacking, respectively (Table 1). The statistical analyses shown in Figure 1 show that the improvement in the HFRPC is statistically significant for base classifiers LR and KNN in terms of all four metrics.

However, this improvement was not significant in the other base classifiers and all other ensemble methods. The ROC comparisons are shown in Figure 2.

**Table 1.** Performance comparisons between the HFRPC with each base classifier and four benchmark ensemble models for HNSCC survival prediction. The best results are marked in bold.

| Models | | ACC | AUC | SEN | SPE |
|---|---|---|---|---|---|
| **Base classifiers** | GB | 0.715 | 0.830 | 0.930 | 0.532 |
| | LR | 0.712 | 0.762 | 0.651 | 0.765 |
| | QDA | 0.638 | 0.681 | 0.343 | 0.889 |
| | KNN | 0.644 | 0.640 | 0.673 | 0.619 |
| | DeT | 0.879 | 0.873 | 0.902 | 0.859 |
| | RF | 0.801 | 0.881 | 0.819 | 0.786 |
| | XGBoost | 0.816 | 0.827 | **0.968** | 0.686 |
| **Ensemble methods** | PV | 0.851 | 0.912 | 0.892 | 0.816 |
| | WF | 0.882 | 0.910 | 0.889 | 0.876 |
| | Stacking | 0.876 | 0.912 | 0.819 | **0.924** |
| | DT | 0.882 | 0.915 | 0.895 | 0.870 |
| **HFRPC** | | **0.885** | **0.930** | 0.892 | 0.878 |

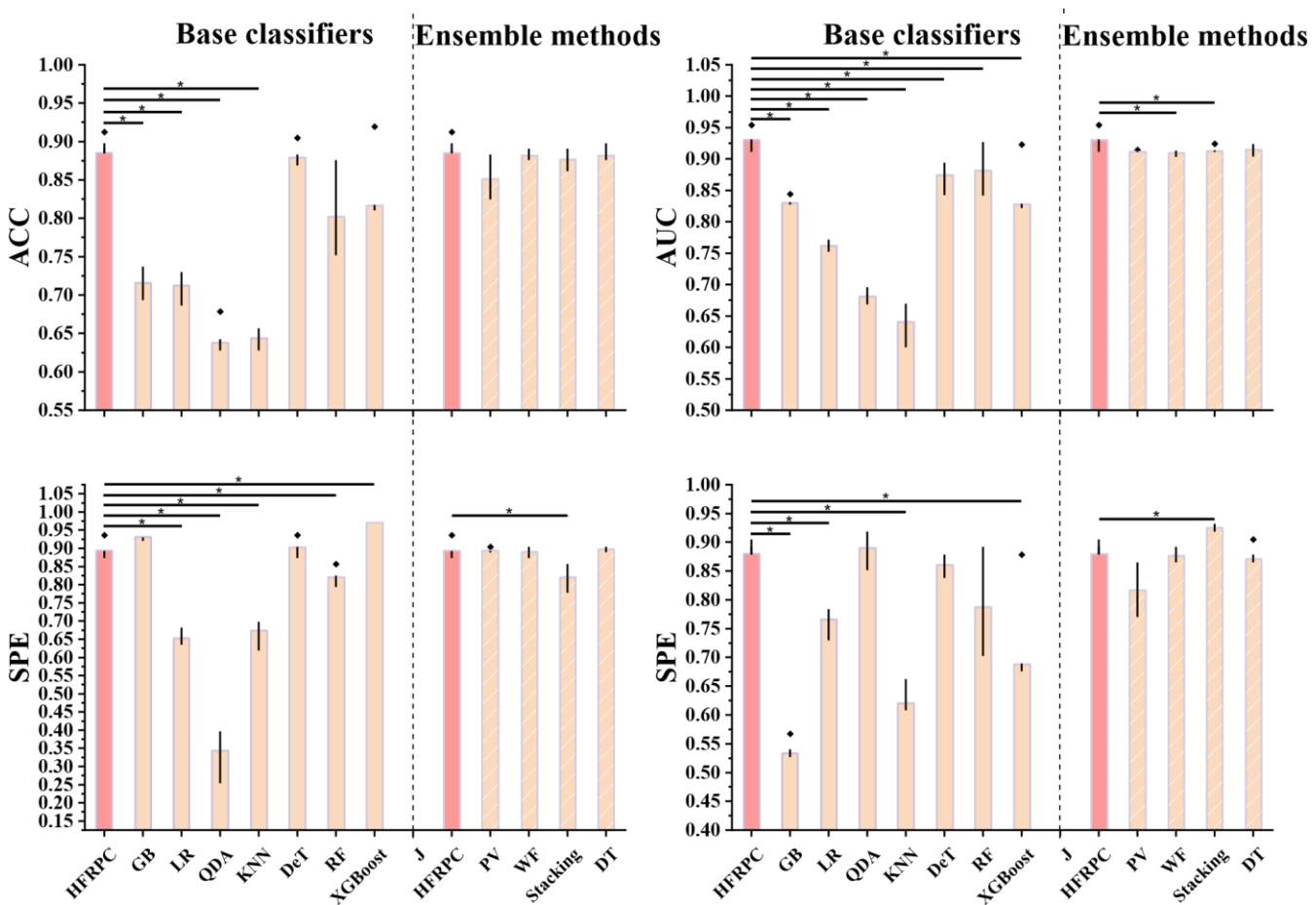

**Figure 1.** Statistical analyses (Wilcoxon signed rank test) between the HFRPC and base classifiers and the benchmark ensemble method survival prediction for HNSCC radiation therapy patients. Asterisk '*' marks are provided between two models with significant difference ($p < 0.05$).

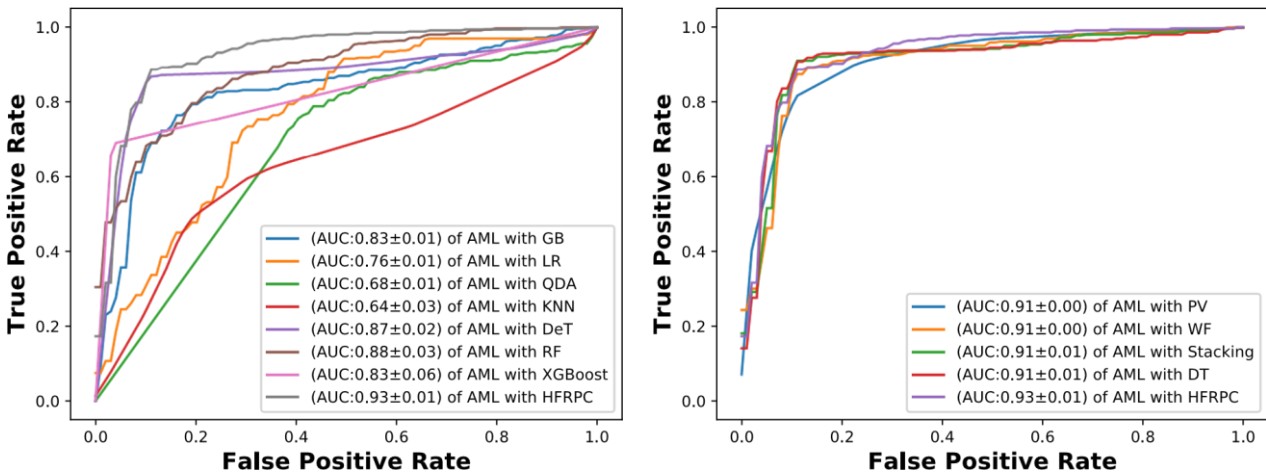

**Figure 2.** ROC analyses between the HFRPC with the (left) base classifiers and (right) benchmark ensemble methods in HNSCC survival prediction.

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
