# OpenReview forum: "A hierarchical fusion framework integrating random projection-based classifiers: application in head and neck squamous carcinoma cancer"
_MIDL.io/2020/Conference — Submitted to MIDL 2020_

### Official Review · AnonReviewer1 · 2020-03-10
**Ensemble fusion approach for combining classifiers**

**Rating:** 1
**Confidence:** 5

**Review:**

- Introduction is a summary of ensemble learning, rather than what the problem is with existing methods that needs to be solved. This means the solution is not well motivated or clear as to the reasoning.
- Weird formatting issues in the paper, unclear notation.
- Not clear what the different ensemble methods are
- Clinical problem and dataset not described, not clear how classes were defined.

---

### Official Review · AnonReviewer3 · 2020-03-11
**More metholodgy and experiment details needed**

**Rating:** 2
**Confidence:** 4

**Review:**

Authors proposed a hierarchical fusion framework to integrate random classifiers. It is an interesting work and could have clinical impacts. Experiments show a large improvements from each individual classifier.

Major concerns:

 - Methods section is too short for readers to really understand HFRPC. It is necessary to summarize the two-level using equations. How to choose weights for each base classifier ? Also, Eq 1 is not clear due to the format issue.

 - Experiment section lacks lot of details. what value you are going to predict in survival prediction ? Did you predict if the patient can survival more than 2 years ? Also, how many radiomics features are extracted and did you use feature selection ?
In survival prediction, it usually reports C-index or AUC if the target survival year is specified.

 - Technical novelty is limited.

---

### Official Review · AnonReviewer4 · 2020-03-11
**A classifier ensembling strategy is proposed and evaluated for prediction of survival in head&neck cancer patients**

**Rating:** 1
**Confidence:** 4

**Review:**

This short paper proposes a two-level ensembling strategy, and evaluates its performance on a dataset of 137 patients with head&neck cancer, for predicting survival (yes/no). As input features, radiomics features and clinical features are combined.
Strengths:
- The proposed ensembling strategy is compared with several other ensembling methods and with individual classifiers.
- The results indicate competive performance of the proposed method.
Weaknesses:
- Unfortunately, the description of the method is too short and vague to assess its theoretical validity, novelty, and relation to existing methods. It is also unclear whether any hyperparameters are involved in this new method, and how they were tuned. The results are encouraging, but the lack of clarity about the method is a major concern.
Details:
- There seems to be a corrupt equation at the end of Section II.
- Fig 2: what is "AML"?

---

### Official Review · AnonReviewer2 · 2020-03-13
**The paper presents and ensemble method for carcinoma cancer classification. Valid paper. Does not pass threshold.**

**Rating:** 2
**Confidence:** 4

**Review:**

The idea is valid. And random projections are one method for stratified bootstrap sampling of datasets. Technically correct bu this paper falls significantly below the threshold of this conference. It is a passable workshop paper. The significance is rather weak.

The evaluations reported in the table are the same as in figure. Putting both is redundant.
The presentation can be improved a lot.
There are meager significant differences of proposal with existing baseline ensembles.

---

### Meta-Review · Area_Chair1 · 2020-04-06
**MetaReview of Paper19 by AreaChair1**

**Rating:** 1

**Metareview:**

The paper presents an ensemble method for carcinoma cancer classification.

Unfortunately, the description of the method is too short and vague to assess its theoretical validity, novelty, and relation to existing methods.



**Paper Type:**

methodological development

---

### Decision · Program_Chairs · 2020-04-11

Reject